# Scaled-Up Multi-Needle Electrospinning Process Using Parallel Plate Auxiliary Electrodes

**DOI:** 10.3390/nano12081356

**Published:** 2022-04-15

**Authors:** Étienne J. Beaudoin, Maurício M. Kubaski, Mazen Samara, Ricardo J. Zednik, Nicole R. Demarquette

**Affiliations:** Department of Mechanical Engineering, École de Technologie Supérieure, Montréal, QC H3C 1K3, Canada; etienne.beaudoin.1@ens.etsmtl.ca (É.J.B.); mauricio.kubaski.1@ens.etsmtl.ca (M.M.K.); mazen.samara@etsmtl.ca (M.S.); ricardo.zednik@etsmtl.ca (R.J.Z.)

**Keywords:** electrospinning, multi-needle electrospinning, scale-up, auxiliary electrodes, nanofibers, non-woven membranes, PVDF

## Abstract

Electrospinning has gained much attention in recent years due to its ability to easily produce high-quality polymeric nanofibers. However, electrospinning suffers from limited production capacity and a method to readily scale up this process is needed. One obvious approach includes the use of multiple electrospinning needles operating in parallel. Nonetheless, such an implementation has remained elusive, partly due to the uneven electric field distribution resulting from the Coulombic repulsion between the charged jets and needles. In this work, the uniformization of the electric field was performed for a linear array of twenty electrospinning needles using lateral charged plates as auxiliary electrodes. The effect of the auxiliary electrodes was characterized by investigating the semi-vertical angle of the spun jets, the deposition area and diameter of the fibers, as well as the thickness of the produced membranes. Finite element simulation was also used to analyze the impact of the auxiliary electrodes on the electric field intensity below each needle. Implementing parallel lateral plates as auxiliary electrodes was shown to help achieve uniformization of the electric field, the semi-vertical angle of the spun jet, and the deposition area of the fibers for the multi-needle electrospinning process. The high-quality morphology of the polymer nanofibers obtained by this improved process was confirmed by scanning electron microscopy (SEM). These findings help resolve one of the primary challenges that have plagued the large-scale industrial adoption of this exciting polymer processing technique.

## 1. Introduction

Polymeric nanofibers are seeing increased application in a wide range of fields, ranging from drug delivery to sensors and advanced filtration [1]. In many cases, the polymeric nanofibers are produced using the electrospinning technique [2,3,4], where a single needle produces fibers at a rate of 0.01 g/h to 2 g/h of deposited polymer [5]. This low production yield leads to high production costs and precludes industrial adoption of this scientifically important technique. The growing urgency to scale-up the electrospinning process has led to the investigation of needleless electrospinning, which has shown to be able to quickly produce nanofibers in large quantity [6,7]. This technique, although faster, suffers from drawbacks compared with needle-based electrospinning, including wider fiber diameter distribution, the solution bath being exposed to air resulting in premature solvent evaporation, and difficulty in producing core-shell fibers [7]. Another way to scale-up the electrospinning process is to use multiple needles, which allows a fast, and perhaps simpler scale-up than needleless electrospinning [8]. This implementation, however, suffers from poor production quality due to Coulombic repulsion between the charged jets and needles that leads to an uneven electric field distribution along the multiple needle array. As the electric field is the driving force pulling and stretching the polymeric jets in electrospinning [3], this instability in the electric field precludes the production of high-quality fibers during multi-needle electrospinning process.

Numerous researchers have tried to address these electric field issues using varying needle arrangements [9] and needle spacing [10], adding a dielectric material around the needles [11], changing needle to collector distances [12], complementing coaxial needles with compressed air flow [12,13], and even applying a different voltage to each needle [10]. Other approaches include the introduction of auxiliary electrodes placed perpendicularly to the needles [14], cylindrical electrodes surrounding the needles [15], and even ring-shaped electrodes [16,17] in attempts to improve the quality of deposited fibers. Despite considerable effort, none of these embellishments has been able to yield the required uniformization of the electric field.

In the present work, a linear array of twenty needles was used to scale up the electrospinning process. As expected, the electrospinning jets were found to interact with one another, resulting in non-uniform fiber diameter and membrane thickness. For most applications, the uniformity of the fiber diameter and the thickness of the deposited non-woven fiber membrane are critical parameters determining the quality of the material produced [18,19]. We found that these issues could be resolved by implementing charged lateral plates as auxiliary electrodes. The improvements were confirmed experimentally and the uniformization of the electric field was established by running a finite element simulation. This optimized technique was found to quickly produce large areas of high-quality nonwoven polymer nanofiber membranes.

## 2. Materials and Methods

Polyvinylidene difluoride (PVDF) powder from Arkema (King of Prussia, PA, USA) with the trade name Kynar 741 was dissolved in a solvent composed of 70% n,n-dimethylformamide (DMF) and 30% acetone (by weight, from Fisher Scientific Company, Fair Lawn, NJ, USA), to obtain an 18%wt PVDF solution, as reported in a previous work [20]. The mixing of the solution was performed with a magnetic stirrer on a hot plate at 70 °C for 4 h. This was followed by a 30-min cool down to room temperature under continued stirring. The electrospinning was carried out using a FLUIDNATEK L-100 machine by Bionicia (Paterna, Valencia, Spain). A linear array of twenty needles, positioned 13 mm apart was used. The solution flowrate through each needle was 0.5 mL/h. The applied voltage to the needles was 15 kV and the needles were placed at 20 cm from the collector to which −15 kV was applied. A simple flat collector and a conveyor belt collector, with travel speed of 0.5 mm/s, were used in this work. The fibers were collected on an aluminum foil substrate. A micrometer was used to measure the thickness of the membranes, the average of 5 measurements is reported, and the standard deviation is represented as the error bars. Disposable 21 gauge needles were used.

A Hitachi S3600 (Hitachi, Ltd., Tokyo, Japan) scanning electron microscope (SEM) was used to assess the fibers. ImageJ software 1.53a (Bethesda, Rockville, MD, USA) was used to measure the diameter of the fibers; 100 measurements were made per sample on three different images and the standard deviation is reported.

COMSOL Multiphysics 5.6 (Burlington, MA, USA) was used to model the intensity of the electric field just below the needles. To do so, both Gauss’ and Faraday’s Laws (Equations (1) and (2)) were solved using the electrostatic stationary physics interface. In these equations, **D** is the electric displacement field, $\rho_{v}$ is the volumetric charge density, **E** is the electric field, and *V* is the electric potential. The minimum and maximum free tetrahedral element size was 0.003 and 1 mm, respectively. In the simulation, stainless steel chromium steel, from the COMSOL library, was used as the material for the needles, air for the space domain, and pure aluminum for the collector.
(1)$$\nabla \cdot D=\rho_{v}$$
(2)E=−∇V

## 3. Experimental Results

A representative in situ image showing the multi-needle electrospinning apparatus is shown in Figure 1. All 20 needles making up the linear array were used simultaneously in parallel during fiber production.

As indicated by the orange arrows in Figure 1, the spun jets from the side needles were deflected by around 45° from the vertical. The consequence of this deflection can be seen in the deposition pattern of the spun fibers, as presented in Figure 2.

The deposition pattern shows a well-defined middle section, where the deposition spots are uniform in size and density, and two distinct side sections, where the deposition spots are spread over a wider area, which decreases the deposition density of the fibers.

The resulting fibers were analyzed by SEM and representative images of the fibers are shown in Figure 3. It can be observed that the fibers in the middle section have a significantly larger diameter than those in the side sections, an average of 445 nm and 288 nm, respectively.

Using a conveyor belt as a collector, large membranes were produced, and the thickness was measured along the width, as presented in Figure 4.

These deposition non-uniformities are due to electric field interaction between the needles. A theoretical explanation for this phenomenon is presented in the discussion section.

This issue was resolved using parallel plate auxiliary electrodes: a folded aluminum foil sheet was positioned on each side of the needle array, both connected to the same voltage source as the needles (15 kV with −15 kV at the collector). This improved setup is presented in Figure 5, where the parallel plate auxiliary electrodes are shown within the red circles.

Figure 5 also shows, as indicated by the orange arrows, that when parallel plate auxiliary electrodes are used, no deflection of the electrospinning jets is observed. This results in a greatly improved, uniform deposition pattern (Figure 6).

After the introduction of the parallel plate auxiliary electrodes, the deposition pattern no longer shows distinctive middle and side sections. It is at this point relevant to mention that, since most scaled-up electrospinning systems use either a conveyor belt or a rotative drum as the collector, fiber deposition will be uniform over both the width and the length of the produced membranes, due to the motion of the collector. Representative SEM images are shown in Figure 7, confirming that the diameters of the produced fibers no longer vary significantly across the deposition width.

Using this optimized fiber-producing technique that includes the parallel plate auxiliary electrodes, large nonwoven polymer nanofiber membranes were electrospun using a conveyor as the collector, as shown in Figure 8. The thickness profile along the width of the membrane, presented in Figure 9, shows that fibers were contained within a narrower span and uniform thickness was obtained.

## 4. Discussion

To better understand the experimental observations, the electric field intensity across the needle array was computed using COMSOL. To do so, a 3D geometry was used, as presented in Figure 10. Cylinders were used to model the needles, as presented in the geometry shown in isometric perspective, Figure 10. The same applied voltage, inter-needle distance, and needle-to-collector distance were used as for the experiment, which is to say 30 kV, 13 mm, and 20 cm, respectively. An air domain of 400 × 150 × 250 mm (width × depth × height) was used to surround the needles and the collector. The size of the collector was 350 × 100 × 10 mm. The electric field was calculated 1 mm below the array of needles, consistent to what is customarily reported in the literature [9,11]. The results of the simulation are presented in Figure 11, where each peak corresponds to a needle.

Perhaps counterintuitively, the electric field near each needle depends greatly on the relative position of the needle along the needle array, as illustrated in Figure 11. Indeed, the model calculations confirm that the magnitude of the electric field 1 mm directly below the needle is approximately 50% greater for the needles located at the extremes of the needle array compared with the needles at the center of the array. This non-uniformity of the electric field is due to the interaction between needles: each needle acting as a positively charged electrode, imposing an electric field on its neighbor. A needle located in the center of the array will undergo the effect of these imposed electric fields from a similar number of needles on the right and left sides; the sum of these imposed electric fields of opposite directions will then cancel each other out. The situation will be different for needles located far from the center: in this case, the electric fields imposed from the neighboring needles on either side of a given needle will no longer cancel out, resulting in a net electric field greater than zero. This will, along with the electrical field toward the collector, result in an electric field of higher total magnitude.

This non-uniformity of the electric field due to the interaction between the needles explains the non-uniformities observed experimentally: indeed, during the electrospinning process, the electric field is responsible for pulling the polymeric jets out from the needles. An inhomogeneous electric field will, therefore, result in unstable jets pulled at a deflected angle: the jets on the sides are pushed outward and are more dispersed, unlike the jets in the middle, that are constrained by their nearest neighbors.

It is well known in the literature that a stronger electric field causes a stronger stretching of the polymeric jet, resulting in smaller fiber diameters [21]. This explains why, in the unmodified multi-needle apparatus that lacks the auxiliary electrodes, the diameter of the fibers deposited away from the center of the needle array is smaller than of those deposited in the middle.

To resolve these issues, parallel plate auxiliary electrodes were implemented, as presented in the experimental section. This improved setup was modeled by adding lateral conducting plates that were subjected to the same voltage as the needles, as shown in the new geometry presented in Figure 12. The electric field was also computed 1 mm below the needle array, and the results are presented in Figure 13, where the peaks corresponding to the parallel plate auxiliary electrodes are shown within the red circles.

As illustrated in Figure 13, an almost perfectly uniform electric field distribution can be obtained using this parallel plate auxiliary electrode geometry. This uniformization can be explained by considering that the lateral auxiliary electrodes impose an additional electric field on the needles. If the strength of the imposed electric field coming from these auxiliary electrodes is equal to the strength of the imposed electric field coming from the neighboring needles, both sets of electric fields will cancel each other out. The strength and direction of the electric field imposed by the auxiliary electrodes can be optimized by changing the distance between the auxiliary electrodes and the needles, and by changing the auxiliary electrode geometry. Finite element modeling shows that a uniformly distributed electric field along the needle array can be achieved by using parallel auxiliary electrode plates of 20 × 20 mm placed at a distance of 16 mm from the nearest needles.

Experimentally, it was found that the auxiliary electrode plates had to be made a little longer (approximatively 3.5 cm) to compensate for the Coulombic repulsion of the charged jets. Since any electrical charge on the polymeric jets was unaccounted for in the equilibrium-state finite element model, this was to be expected.

When properly optimized, the lateral auxiliary electrode plates repel the side jets with the same strength as the other needles, realigning the jets toward the collector and eliminating the deflection angle. Since all the electrospinning jets are now vertically oriented directly toward the collector, the fibers are deposited uniformly. We, therefore, find that the uniformity of the electric field obtained across the needles using parallel auxiliary electrode plates creates the necessary stable processing conditions for the production of high-quality fibers of homogeneous diameter.

Parallel plate auxiliary electrodes have, therefore, proven to be a simple and efficient way to optimize the multi-needle electrospinning process. The simulation results were found to be in agreement with the experimental results. Since no semi-vertical jets were observed, a uniform deposition pattern was produced. Since fiber diameter along the needle array was found to be uniform, it can be, reasonably, concluded that the electric field is quite uniform, which indicates steady processing conditions.

## 5. Conclusions

The use of a multi-needle apparatus seems an obvious way to scale up the electrospinning process to increase production yields. However, as numerous researchers have found, such an implementation suffers from unstable processing parameters due to the non-uniformity of the electric field, resulting in the deflection of electrospinning jets that cause heterogenous fiber deposition and membrane thickness, as well as poor fiber diameter control.

We found that this challenge can be readily overcome by introducing parallel auxiliary electrode plates that confine the needle array. Experimental observations and finite element computational models confirm that the introduction of these additional electrodes achieves uniformity of the electric field along the needle array. This greatly improves the process parameter controls, thereby enabling the scaled-up high-yield production of good quality, homogeneous polymeric nanofibers using the electrospinning technique.

## Figures and Tables

**Figure 1 nanomaterials-12-01356-f001:**
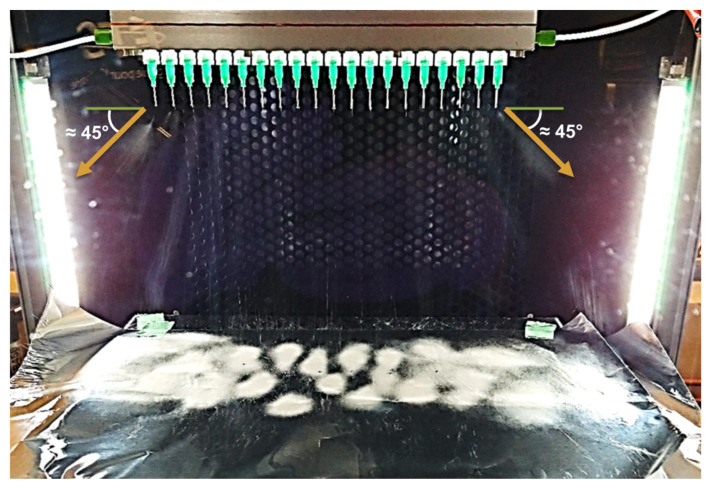
Multi-needle electrospinning, initial setup (without auxiliary electrodes).

**Figure 2 nanomaterials-12-01356-f002:**
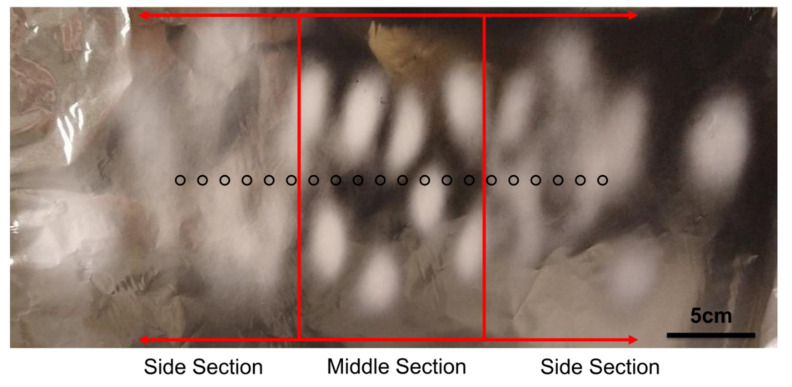
Deposition pattern of the initial setup (without auxiliary electrodes). The black circles represent needle position.

**Figure 3 nanomaterials-12-01356-f003:**
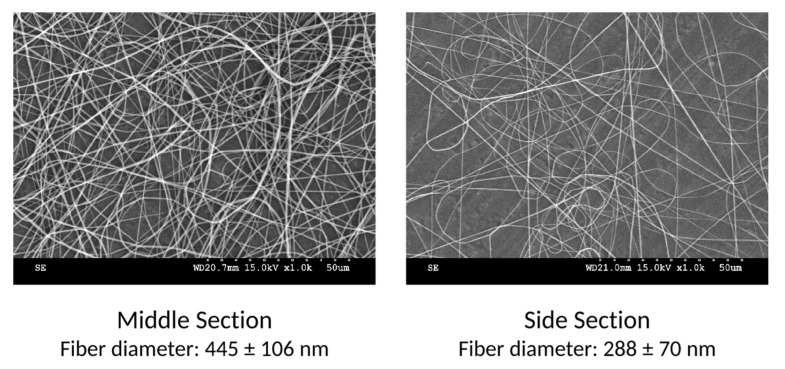
Example of the fibers from the middle and the side sections with initial setup (without auxiliary electrodes).

**Figure 4 nanomaterials-12-01356-f004:**
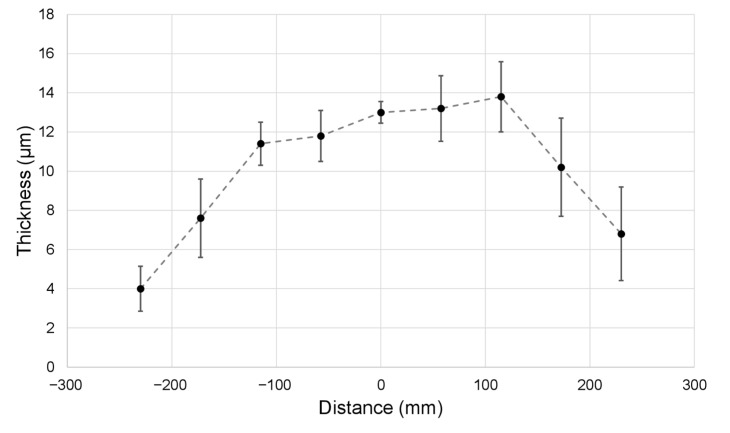
Thickness along the width of the membrane produced with initial setup (without auxiliary electrodes); the dashed line is only a guide for the eyes.

**Figure 5 nanomaterials-12-01356-f005:**
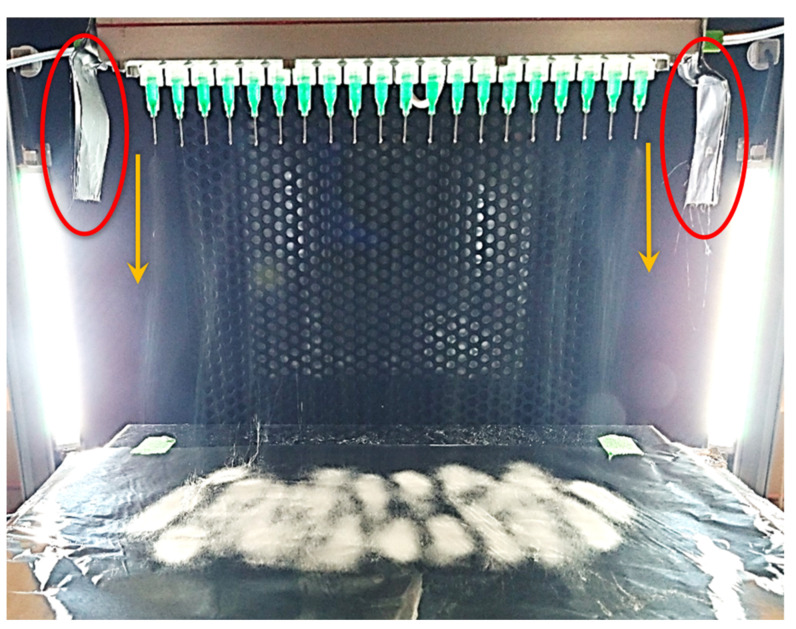
Multi-needle electrospinning with parallel plate auxiliary electrodes (indicated in red).

**Figure 6 nanomaterials-12-01356-f006:**
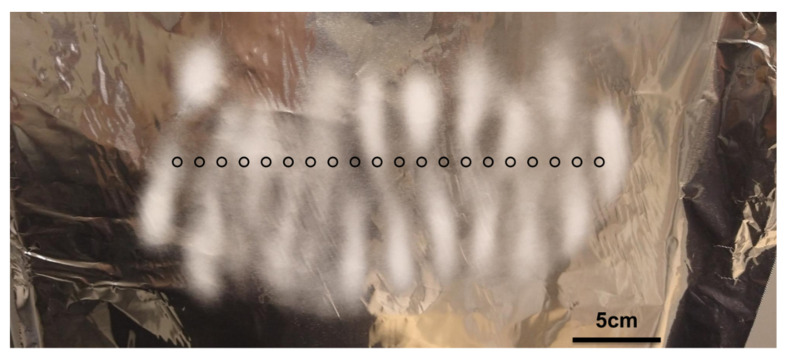
Deposition pattern with parallel plate auxiliary electrodes. Black circles represent needle position.

**Figure 7 nanomaterials-12-01356-f007:**
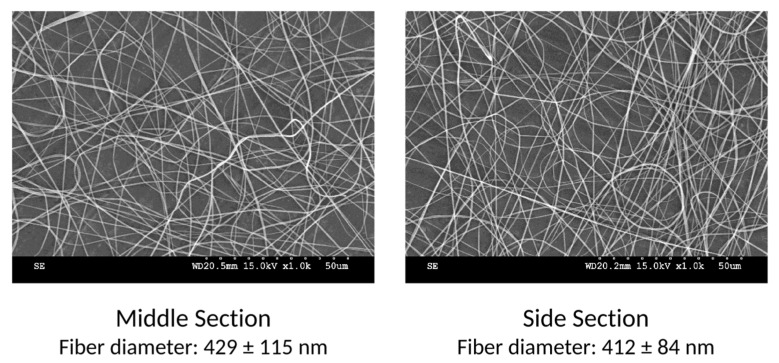
Example of the fibers from the middle and from the side with parallel plate auxiliary electrodes.

**Figure 8 nanomaterials-12-01356-f008:**
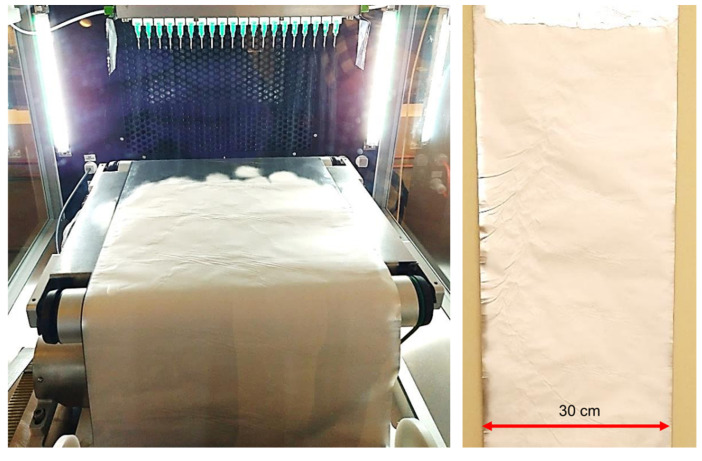
Production of large nonwoven membranes.

**Figure 9 nanomaterials-12-01356-f009:**
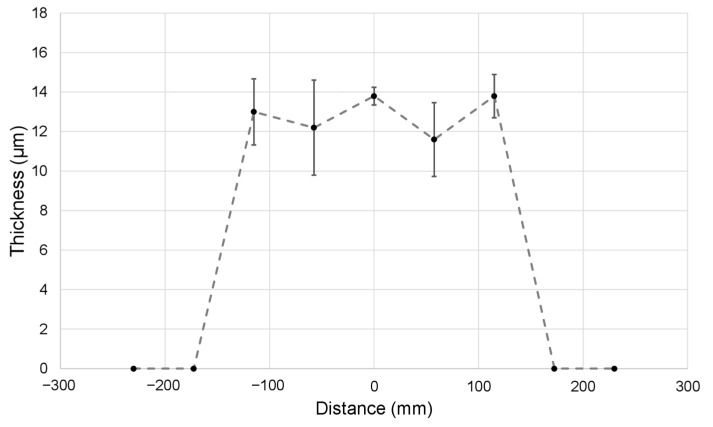
Thickness along the width of the membrane with parallel plate auxiliary electrodes, the dashed line is only a guide for the eyes.

**Figure 10 nanomaterials-12-01356-f010:**
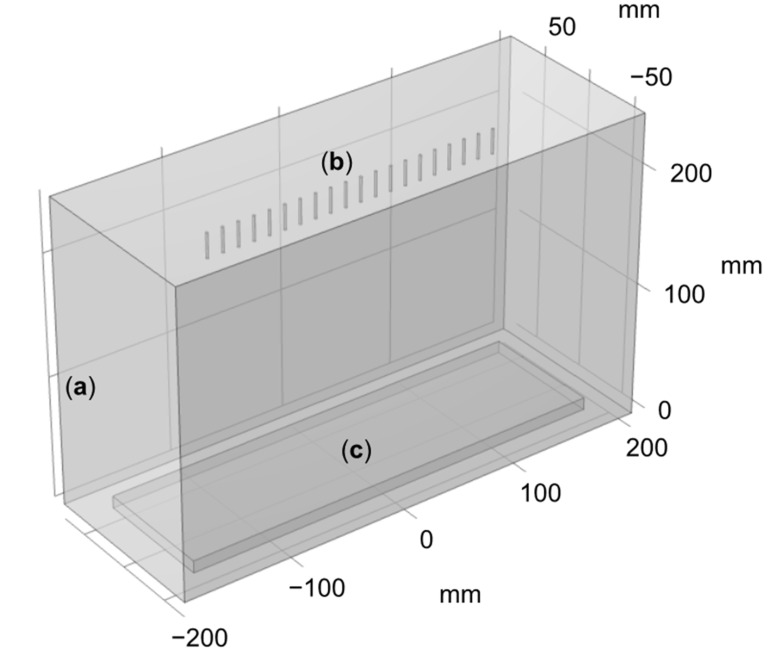
COMSOL 3D geometry, **(a)** air domain, **(b)** needles, (**c**) collector.

**Figure 11 nanomaterials-12-01356-f011:**
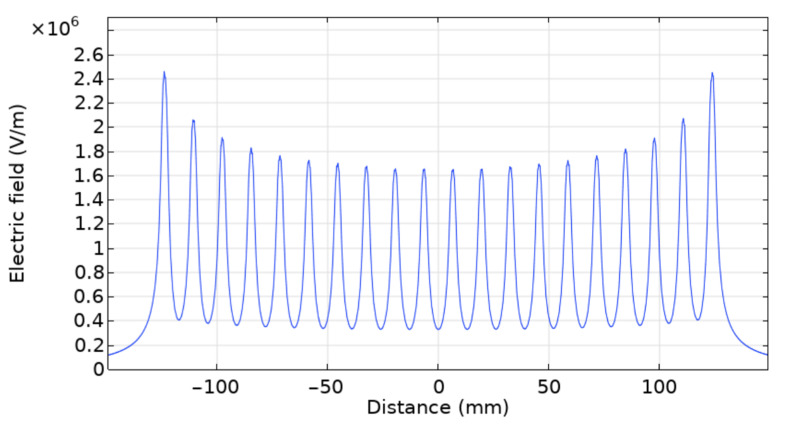
COMSOL electric field simulation initial setup (without auxiliary electrodes), showing the electric field 1 mm below the needle tips.

**Figure 12 nanomaterials-12-01356-f012:**
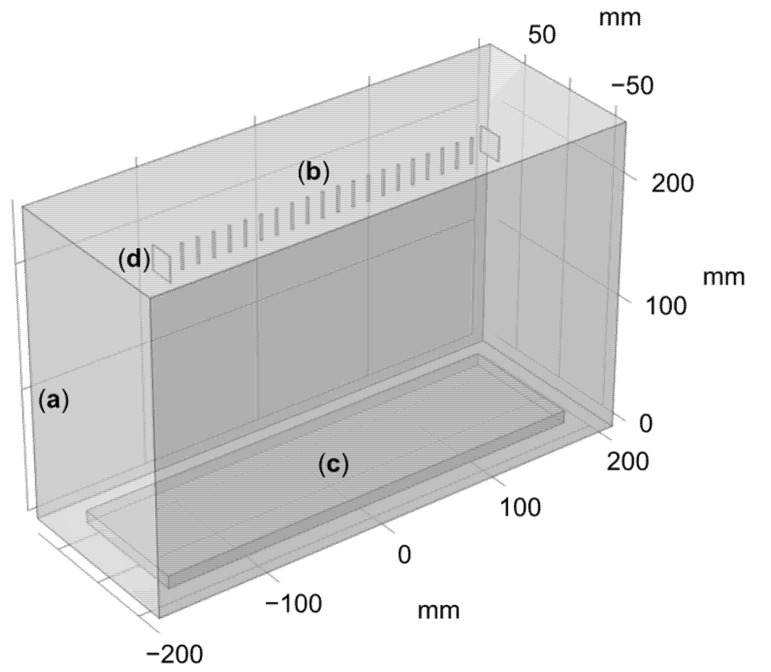
COMSOL Geometry with auxiliary electrodes (**a**) air domain, (**b**) needles, (**c**) collector, (**d**) auxiliary electrodes plate.

**Figure 13 nanomaterials-12-01356-f013:**
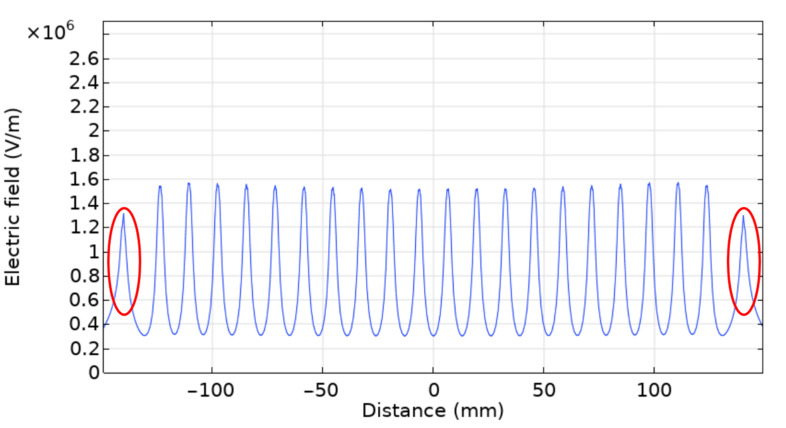
COMSOL electric field simulation with auxiliary electrodes, where the peaks corresponding to the parallel plates auxiliary electrodes are shown within the red circles.

## Data Availability

The data presented in this study are available in this article.

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
