# Peer review of "Scaled-Up Multi-Needle Electrospinning Process Using Parallel Plate Auxiliary Electrodes"

_nanomaterials, 2022, doi:10.3390/nano12081356_

Round 1

Reviewer 1 Report

Manuscript titled: "Scaled-Up Multi-Needle Electrospinning Process Using Parallel Plate Auxiliary Electrodes " is a study on how to overcome the difficulties of electrospinning's up scaling. The aricle is of interest to the readers of NAnomaterials, howvever before submission their are some comments that needs to be addressed and changes to be made.

Comments to the authors:

Introduction:

I am not sure why authors focus only on the electrospinning using needles and do not discuss the advantages and disadvantages, or comparison to large scale needless electtropsinning systems, which are commercially available.

Methodology:

I am not convinced by presented results, whisch suppose 'to prove that fibrous mats obtained by the authors have evenly distributed thickness. Studies over thickness measurements throughout the mat should be provided along with tensile measurements of samples cut out from randomly selected areas of the mat.

Discussion part:

Authors should discuss their results with more recent references.

Author Response

R1 :

Introduction:

I am not sure why authors focus only on the electrospinning using needles and do not discuss the advantages and disadvantages, or comparison to large scale needless electtropsinning systems, which are commercially available.

A section has been added to the introduction, mentioning the scale-up potential of the needle-less electrospinning, and insisting on the potential of multi-needle electrospinning.

Methodology:

I am not convinced by presented results, whisch suppose 'to prove that fibrous mats obtained by the authors have evenly distributed thickness. Studies over thickness measurements throughout the mat should be provided along with tensile measurements of samples cut out from randomly selected areas of the mat.

A study of the thickness of the produced fibrous mats has been conducted and the results have been added to the “experimental results” section, confirming that the use of parallel plates auxiliary electrodes allows the production of uniformly thick fiber mats, compared to those produced with the initial setup.

A study of the tensile strength of the produced membranes is beyond the scope of this research and has not been conducted. The fiber size and membrane thickness are the parameters that are generally considered and therefore should be enough for this scale-up research.

Discussion part:

Authors should discuss their results with more recent references.

No relevant newer articles where found.

Reviewer 2 Report

The present manuscript by Beaudoin et al. describes a simple and efficient way to optimize the multi-needle electrospinning process, which has shown to be one of the effective ways to scale up the electrospinning process. The authors demonstrated experimentally and theoretically that implementing charged lateral plates as auxiliary electrodes brought about the uniformization of the electric field for a linear array of twenty electrospinning needles, resulting in stable processing conditions for the production of non-woven electrospun nanofiber membranes consisted of nanofibers of homogeneous diameter. The manuscript is well written and the data analysis is logical and correct. The topic of this manuscript may attract the readership of Nanomaterials. Therefore, I would recommend its acceptance for publication after addressing following minor issues:

(1) There are 3 Figure 1s. Please combine these 3 Figures.

(2) In Figure 3 and 6, the authors must show SEM images with lower magnification, which were probably used to determine the averaged fiber diameters, additionally to the current images with relatively high magnification because the numbers of the fibers shown in the current SEM images are too small to indicate homogeneity or non-homogeneity of the fiber diameters.

(3) Figure 5 and Figure 11 clearly suggested an almost perfectly uniform electric field distribution in the horizontal direction (“left-right” direction in Figure 5). How about the vertical direction (“up-down” direction in Figure 5)? The authors should discuss this point.

Author Response

R2 :

(1) There are 3 Figure 1s. Please combine these 3 Figures.

It has been corrected.

(2) In Figure 3 and 6, the authors must show SEM images with lower magnification, which were probably used to determine the averaged fiber diameters, additionally to the current images with relatively high magnification because the numbers of the fibers shown in the current SEM images are too small to indicate homogeneity or non-homogeneity of the fiber diameters.

The images have been replaced with new ones that have lower magnification, since it indeed helps to show the homogeneity of the fiber diameters. However, in order to measure precisely the diameter of the fibers, multiple high magnification images have been used.

(3) Figure 5 and Figure 11 clearly suggested an almost perfectly uniform electric field distribution in the horizontal direction (“left-right” direction in Figure 5). How about the vertical direction (“up-down” direction in Figure 5)? The authors should discuss this point.

Since most scaled-up electrospinning systems use either a conveyor belt or a rotative drum as collector, the fiber deposition will, when using the parallel plates and due to the motion of the collector, be uniform over both the width and the length of the produced membranes. This information has been added to the “experimental results” section.